# OpenReview forum: "A Cross-Model Study of Over-Compliance in Large Lan- guage Models"
_TMLR — Under review for TMLR_

### Review · Reviewer_Sd66 · 2026-06-06

**Summary Of Contributions:**

This paper investigates "over-compliance" in large language models, defined as the tendency to generate substantive responses when a prompt is underspecified, ambiguous, contradictory, or nonsensical. The authors construct a benchmark of 400 prompts spanning four categories of input failure and evaluate four frontier models (GPT-4.1-mini, Gemini 2.5 Flash, Llama 3.3 70B Instruct, and Claude Haiku 4.5) under two system-prompt conditions. Responses are classified using a deterministic rule-based classifier consisting of 9 categories, from which Over-Compliance Rate and Terminal Refusal Rate are derived. The study reports substantial variation across models, and argues that prompt-level mitigation is insufficient for reducing over-compliance.


Key strengths:

1.) The authors have taken a good attempt to address an important and underexplored aspect of LLM behavior: Over-compliance, that is distinct from factual correctness and hallucination, and its practical stakes (healthcare, legal, engineering) are convincingly motivated.

2.) They have introduced a nine-category scheme, a methodological contribution that reveals behaviors such as refusal cascade, "clarification plus" versus "pure clarification" that binary schemes obscure.

3.) The cross-model comparison was done empirically with 4 leading providers across 2 conditions * 4 categories * 100 prompts = 3,200 responses.

4.) They shared the striking findings: refusal cascade, Claude at 87% terminal refusal on ambiguity without prompt, dropping to 11% with prompt, with clear deployment implications.

5.) They also shared the source code with an organized README file to reproduce the results.



Key weaknesses:

1.) The central metric depends heavily on a handcrafted rule-based classifier whose validity is only partially established.

2.) The rule-based classifier relies on surface patterns such as bullet points, "in the meantime", medical disclaimers). The authors report Cohen's kappa = 0.725 against a 112-row human sample, but per-model kappa drops to 0.54 for GPT-4.1-mini, attributed to Pure Clarification concentration. This suggests systematic misclassification risk, particularly for subtle or novel refusal idioms.

3.) The authors acknowledge that explaining why a contradiction is impossible may be misclassified as over-compliant. This is not a minor caveat; it cuts to the validity of the OCR metric for those categories. A response like "You asked for a number that is both even and odd. That is impossible because…" is arguably appropriate engagement, not over-compliance.

4.) Statistical rigor is limited because each experiment is performed only once, despite stochastic generation settings. For example, the refusal cascade might be stochastic or sensitive to prompt ordering within the 100-prompt thread.

5.) The sequential prompting design introduces substantial context effects, making it difficult to separate model behavior from conversation-history artifacts. Also, placing 100 prompts in a single thread per category means that later responses are influenced by earlier ones. The refusal cascade is both a finding and a design artifact. It is unclear whether the same cascade would appear with interleaved categories or shorter threads.

6.) The minimal prompt, such as "Respond naturally. If unclear, ask for clarification." is a weak intervention. The conclusion that "prompt-level mitigation is unreliable" is overstated given that more sophisticated prompting, chain-of-thought refusal, explicit contradiction detection was not tested.

7.) The authors should have done a direct comparison against alternative scoring frameworks or human judgments at scale.

8.) Now there are so many latest models like GPT-5, 5.1, Claude Sonnet 4.6. It would have been better to incorporate these models to support the findings of the paper.

**Audience:**

Yes

**Audience Explanation:**

Those who are working on LLM evaluation, alignment, safety, and deployment would find this valuable:

1.) Deployment engineers choosing models for customer-facing applications where input quality is uncontrolled. The finding that GPT-4.1-mini with a clarification prompt reduces OCR to 31% on underspecification and 3-4% on ambiguity is relevant.

2.) The nine-category taxonomy and the demonstration that binary scoring hides critical refusal patterns are a good methodological contribution.

3.) The refusal cascade and the cross-model variability suggest that response policies are shaped by training decisions (RLHF reward structures, sycophancy mitigation) in ways that are not yet well understood.

4.) The paper provides a concrete, replicable protocol for assessing whether a model "knows when not to answer," which is relevant to emerging AI governance frameworks.

**Broader Impact Concerns:**

I do not see major ethical concerns requiring substantial additional broader-impact discussion. The paper already includes a broader impact statement, and it is appropriate and responsible.

**Claims And Evidence:**

Yes

**Claims Explanation:**

The paper provides evidence supporting the existence of model-dependent differences in response behavior on underspecified and ambiguous prompts. The qualitative examples and reported OCR and TRR differences are generally consistent with the paper's central claims. However, the evidence is not fully sufficient to support the stronger quantitative conclusions. Several points introduce uncertainty:

1.) The entire analysis depends on a deterministic rule-based classifier. While the authors report Cohen's kappa of 0.725 against a 112-sample human-labeled subset, this validation set is relatively small compared to the 3,200 evaluated responses. The paper also acknowledges that classifier errors occurred and required manual correction.

2. The definition of over-compliance appears problematic for contradiction and nonsense categories. The paper itself notes that responses explaining why a contradiction is impossible are still counted as over-compliant. Consequently, the reported near 100% OCR rates for these categories may partly reflect limitations of the taxonomy rather than true model failure.

3. Experiments were conducted with a single run per model-condition pair despite temperature being set to 1.0. No confidence intervals, variance estimates, or significance testing are reported.

4. Prompts were submitted sequentially within the same conversation thread. The observed Claude refusal cascade may therefore be partly attributable to accumulated conversational state rather than prompt-specific behavior.

**Requested Changes:**

The discussion of the following points would strengthen the paper.

1.) Address the contradiction and nonsense classification problem more rigorously. The current classifier labels any substantive response as over-compliant, including responses that correctly identify logical impossibility or category errors. This inflates OCR in a way that does not represent harmful behavior.

2.) Add statistical uncertainty quantification.

3.) Disentangle the sequential submission confound.

4.) Also, expand the human validation sample and add a discussion of practical mitigation strategies beyond prompting.

---

> ### Author Response · Authors · 2026-07-01
> **Response to Reviewer Sd66**
>
> Thank you for the thorough and supportive review. The four requested changes are addressed below.
>
> **Contradiction and nonsense classification.** This is addressed in two ways. First, a tenth category, Impossibility Flag, was introduced for responses that explicitly state the request cannot be satisfied and decline to produce the artifact, and all 1,200 contradiction and nonsense responses with full text available were re-scored; only 2 of 1,200 moved from over-compliant to clean, and adjusted over-compliance differed by at most 0.5 percentage points per cell, so the near-ceiling rates are not an artifact of penalising constructive impossibility-handling. Second, a strict Over-Compliance Rate is now reported alongside the broad one; the strict definition treats impossibility explanations and generic engagement as clean and counts only responses that commit to answering. Over-compliance remains pervasive (66 to 92 percent) under the strict definition, so the central conclusion is robust to the definitional choice.
>
> **Statistical uncertainty quantification.** The ambiguity category was re-run for all four models with three seeded runs per condition, and means are now reported with 95 percent confidence intervals. The interval half-widths are narrow (0.7 to 6.8 percentage points), supporting the stability of the corrected rates.
>
> **Sequential submission confound.** The independent-session re-run isolates this confound. Under fresh threads Claude's Terminal Refusal Rate on ambiguity falls from 87 percent to 0 percent across all three runs, showing the cascade is an emergent consequence of accumulated conversational state rather than a stable per-prompt policy; the same confound inflated Llama's over-compliance, which falls from 99 percent to 19 percent without the system prompt. Independent-session evaluation is now the default for the corrected analysis.
>
> **Expanded human validation and mitigation beyond prompting.** The human validation was enlarged from 112 to 315 labelled responses, stratified by taxonomy category, with per-category precision and recall now reported. This is stated transparently: fine ten-category agreement is Cohen's kappa 0.36 and binary over-compliance agreement is 0.58 after a content-based revision of the classifier, both lower than the 0.725 originally reported on the smaller input-balanced sample, which was a harder and fairer test. On mitigation, a paragraph was added noting that, beyond prompting, mitigation may require an external input-validation layer, targeted fine-tuning on clarification and abstention, uncertainty-gated deferral that withholds an answer when confidence signals are low, or reward-model adjustment that reinforces appropriate deferral.
>
> **On newer models (weakness 8).** This is acknowledged as a limitation and added as future work: the benchmark, taxonomy, and corrected protocol are designed to be directly replicable, and a natural next study would run all four input-failure categories under the corrected independent-session protocol on the latest model generation to test whether response policy has improved and whether the cross-model differences persist.

---

> ### Comment · Reviewer_Sd66 · 2026-07-04
>
> Thanks for clarifying all concerns.

---

### Review · Reviewer_PuUR · 2026-06-06

**Summary Of Contributions:**

This paper studies the ability of medium scale LLMs to respond to poorly formed input across multiple categories (underspecification, ambiguity, contradiction, nonsense).  It showed issues with all 4 models it tested, with some variation within models, and little effect from a simple system prompt.

**Audience:**

Yes

**Audience Explanation:**

The study and testing of language models and their capabilities is of central interest to the ML community.  While this is a snap-shot study of what is the state of tech *now*, it identifies a pressing issue, and has potential to help move the field forward in the short term.

There is a nice related work section that does highlight similar work in this domain.  Some prior work (mostly pre-prints) does tackle a subset of these issues.  This paper gives and an updated view, and a more general evaluation across more fine-grained categories.

The paper does not provide an (anonymized) link to code or full prompts, which limits reproducibility, use, and interest of the readership.

**Claims And Evidence:**

Yes

**Claims Explanation:**

The study is well-described, and experimental results are presented clearly and analyzed fairly.  Results are not over-stated.

**Requested Changes:**

- can you provide access to your code or at least the explicit set of prompts/test-data for this evaluation.

 -  Some of the ambiguous question examples "What’s the best way to invest money?" seems somewhat reasonable to me.  I imagine an LLM can be useful be providing a response which provides options based on various risk tolerance and time horizons.  For instance, there are surely articles on the web about this, and as we transition from a key-word based search world (from the past 20-30 years) to an LLM-based one, this sort of question seems reasonable.  It would be good to review these prompts, since it may not be as ambiguous as the authors believe.

 - the set-up of putting all questions sequentially in a single thread seems potentially problematic to me.  The refusal cascade result it reveals is one sort-of-interesting outcome.  But that thread may actually add context to questions that are stand-alone out of context.  For instance "How do I fix this error?" may be *not* underspecified if something earlier in the thread could be interpreted as describing an error, or the previous response from the LLM said that the ambiguous input consisted an error in phrasing.
   Can you re-run the ambiguity thread so each question is a new thread?  Or justify why this is not a problem?

 - In Table 3 the "input category" abbreviations were confusing for me.  Can you write out the way to map these abbreviations to the full names (e.g., "Amb. = Ambiguity")

---

> ### Author Response · Authors · 2026-07-01
> ****Response to Reviewer PuUR****
>
> Thank you for the positive assessment and the concrete requests. Each is addressed below.
>
> **Code and prompt availability.** The full benchmark of 400 prompts, the provider-specific runner scripts, the rule-based classifier, all raw model outputs, and the composite and validation artifacts are provided in the (anonymized) supplementary material, and a sentence at the end of the Experimental Procedure now points to this explicitly so the resource is easy to locate.
>
> **Ambiguity prompt review ("What's the best way to invest money?").** This reasoning is correct, and that specific prompt belongs to the underspecification category, not ambiguity; it is precisely a prompt answerable in a hedged, option-spanning way, which is why underspecification is treated as a distinct and milder failure type. On re-inspection, all 100 ambiguity items were confirmed to be referential: each turns on an unbound demonstrative with no antecedent in a single-turn context ("Should I go ahead?", "Is this correct?"), so none can be answered by enumerating options. This distinction is now stated in the benchmark description, with representative examples added to the appendix.
>
> **Sequential single-thread confound.** The ambiguity category was re-run with each prompt in an independent, single-turn session, across all four models and three seeded runs. The concern is borne out: under fresh threads Claude's terminal-refusal cascade disappears entirely (87 percent to 0 percent) and Llama's over-compliance falls from 99 percent to 19 percent, confirming that accumulated context, not stand-alone behaviour, drove part of the original result. The corrected analysis is now based on the independent-session runs, with the sequential results retained only as a separately labelled context-accumulation phenomenon.
>
> **Table 3 abbreviations.** The caption now states the mapping explicitly: Und. = Underspecification, Amb. = Ambiguity, Con. = Contradiction, Non. = Nonsense.

---

> > ### Comment · Reviewer_PuUR · 2026-07-01
> >
> > Thanks for the updates and response.  This addresses my concerns, which were relatively minor to begin with.

---

### Review · Reviewer_tD4b · 2026-06-19

**Summary Of Contributions:**

**Summary**
This paper examines over-compliance in large language models, where a model gives substantive answers to prompts that are underspecified, ambiguous, contradictory, or semantically incoherent, instead of asking for clarification, refusing, or deferring. The authors build a benchmark of 400 short synthetic prompts across these four failure types and evaluate four representative models from major providers under settings with and without a minimal clarification-oriented system prompt. Responses are classified with a deterministic rule-based classifier based on a nine-category taxonomy, and the paper reports Over-Compliance Rate and Terminal Refusal Rate as the main metrics. The results show clear behavioral differences across models, with overall over-compliance ranging from 58.0% to 98.8%. The clarification instruction changes behavior only modestly, and all models still tend to over-comply on contradictory and nonsensical prompts.

**Strengths**
1. I like that the paper looks at a failure mode that often comes before correctness: a model may answer even when the prompt does not contain enough information. This is a meaningful issue in high-stakes settings, where a fluent answer can make an underspecified query look more actionable than it really is.
2. The notion of over-compliance matches a behavior that many users may have observed in practice. Instead of asking for the missing context, the model often gives a polished but generic response, and treating this as a response-policy problem is a reasonable direction.
3. I also think the taxonomy is a helpful step beyond a simple answer-versus-clarification label. In particular, separating pure clarification from cases where the model gives substantial content before adding a clarifying question is a useful distinction.
4. The cross-model comparison leads to several observations that are worth reporting. These differences suggest that over-compliance is not just a uniform behavior across models.

**Weaknesses**
1. I still find the definition of over-compliance somewhat unclear. Some substantive responses may be appropriate, such as explaining why a contradictory request is impossible or treating a nonsense prompt as a creative request.
2. The study feels closer to a small-scale behavioral audit than a fully rigorous empirical paper. The 400 short synthetic prompts are useful for an initial observation, but they do not show whether the findings hold for real user queries or more natural conversations.
3. I am concerned that the rule-based classifier may be too fragile for the main claims. Since it relies on surface patterns such as length, structure, and refusal phrases, borderline cases may be mislabeled, especially for contradiction and nonsense prompts. The human validation is also limited.
4. The sequential chat-thread protocol introduces a clear confound. Submitting 100 prompts from the same category in one continuing conversation makes the results sensitive to context accumulation, and the Claude refusal cascade may partly reflect this setup rather than a stable model behavior.

**Audience:**

Yes

**Audience Explanation:**

The paper studies a timely and relevant behavior in LLM deployment, namely whether models answer prompts that should instead trigger clarification or deferral. Although the current evidence has limitations, the cross-model observations and response taxonomy may still be useful to readers interested in model reliability and safety, as well as response policy.

**Broader Impact Concerns:**

No additional broader impact concerns. I found this paper already includes a Broader Impact Statement and appropriately notes that the reported over-compliance rates should not be treated as a complete measure of model quality.

**Claims And Evidence:**

No

**Claims Explanation:**

The paper reports interesting observations, but I do not think the evidence is fully convincing for the broader claims. The main results rely on a small synthetic benchmark and a rule-based classifier with limited validation, so the reported over-compliance rates may be sensitive to the benchmark design and classification rules. In addition, the single-run temperature-1.0 setup and sequential chat-thread protocol make the cross-model comparisons and system-prompt effects difficult to interpret robustly.

**Requested Changes:**

1. For contradiction and nonsense prompts, I think the paper should better separate inappropriate content generation from reasonable responses such as pointing out logical impossibility or treating the prompt as creative.
2. Since the main results rely heavily on the rule-based classifier, I would like to see stronger human validation, including a larger labeled sample, category-level precision and recall etc.
3. Given the temperature-1.0 setup, I do not think a single run is enough to support stable cross-model comparisons. Repeated runs, independent-session evaluation, and confidence intervals would make the results more reliable.
4. I also encourage the authors to make the benchmark less artificial by including more realistic prompts, such as real or semi-real user queries, longer prompts and stronger clarification-oriented system prompts.

---

> ### Author Response · Authors · 2026-07-01
> **Response to Reviewer tD4b**
>
> Thank you for the detailed and constructive review. Each requested change is addressed below.
>
> **Contradiction and nonsense: separating inappropriate generation from reasonable responses.** A robustness analysis was added. A tenth category, Impossibility Flag, was introduced for responses that explicitly state the request cannot be satisfied and decline to produce the artifact, and all 1,200 contradiction and nonsense responses with full text available were re-scored. Only 2 of 1,200 moved from over-compliant to clean, and adjusted over-compliance differed from the original by at most 0.5 percentage points per cell, so the near-ceiling rates are not an artifact of penalising constructive impossibility-handling. More directly, a strict Over-Compliance Rate is now reported alongside the broad one; the strict definition treats generic engagement, impossibility explanations, and creative reframes as clean and counts only responses that commit to answering. Over-compliance remains pervasive (66 to 92 percent) under the strict definition.
>
> **Stronger human validation with per-category precision and recall.** The human validation was enlarged from 112 to 315 labelled responses, stratified by taxonomy category so that even rare categories could be estimated, and per-category precision, recall, and support are now reported. This is stated transparently: fine ten-category agreement is Cohen's kappa 0.36, and binary over-compliance agreement is 0.58 after a content-based revision of the classifier, both lower than the 0.725 originally reported on the smaller input-balanced sample. The larger, category-stratified sample is a harder and fairer test. The per-category table shows the classifier is reliable on structurally explicit categories (Pure Clarification and Terminal Refusal precision above 0.85) but weak on the subtle content-versus-clarification boundaries, so the broad rate is treated as an upper bound.
>
> **Repeated runs, independent sessions, and confidence intervals.** The ambiguity category was re-run for all four models with each prompt in an independent, single-turn session, repeated across three seeded runs, and means are now reported with 95 percent confidence intervals. The interval half-widths are narrow (0.7 to 6.8 percentage points), indicating the corrected rates are stable rather than single-run artifacts.
>
> **Sequential-thread confound.** The independent-session re-run confirms the concern directly. Under fresh threads, Claude's Terminal Refusal Rate on ambiguity falls from 87 percent to 0 percent, so the cascade is an emergent consequence of accumulated conversational state rather than a stable per-prompt policy; the same confound inflated Llama's over-compliance, which falls from 99 percent to 19 percent without the system prompt. Independent-session evaluation is now the default for the corrected analysis, and the sequential cascade is re-framed as a separate phenomenon.
>
> **More realistic prompts.** The limitations now state that the synthetic design was chosen for control, that validation against real conversation logs would improve ecological validity, that longer and semantically embedded prompts are future work, and that the mitigation conclusion is scoped only to the single minimal instruction tested; stronger clarification-oriented prompts were not evaluated and might shift behaviour further.
>
> **Transparency note.** The independent-session re-run was conducted for the ambiguity category, where the cascade arose; the other three categories under fresh-thread conditions are flagged in the limitations as a natural next step, alongside replication on newer models.

---

### Review · Reviewer_gLgV · 2026-06-20

**Summary Of Contributions:**

This paper investigates how large language models (LLMs) respond to ill-formed prompts (defined as either incomplete, ambiguous or contradictory). The authors define overcompliance as the production of a significant response instead of a clarification request or refusal. However, moving away from a previous binary categorization between response and refusal, they introduce a finer-grain taxonomy through which to evaluate LLM responses. They assembled a set of 400 short prompts that span four categories of input failure (underspecification, ambiguity, contradiction, nonsense) and compared the behavior of four average-size LLM (allegedly < 100 B) showing variation in overcompliance score across models (58 % to 98.8 %) as well as characteristic behavior for specific models such as Claude Haiku.

**Additional Comments:**

N/A

**Audience:**

Yes

**Audience Explanation:**

The overall point of the paper which is to examine LLM responses to “incorrect” prompts (defined as ambiguous, contradictory, or nonsensical), thus extending the issue beyond ambiguity in vibe coding applications is relevant and compelling as a broad objective.

**Claims And Evidence:**

No

**Claims Explanation:**

The evidence provided is limited by several factors:
- lack of detailed analysis of the underlying mechanisms
- Prompt length and scale of the experiment
- choice of completion parameters in particular Temperature and (when applicable) seed

**Requested Changes:**

While the rationale for departing from a binary classification of (over)compliance is appropriate, there needs to be a better validation of the taxonomy itself. This validation could be either theoretical by a more detailed comparison with the literature (for instance to validate individual items), or empirical, with statistical validation of the ‘metric’ aspects of the taxonomy, being mindful of i) the preservation of ‘even distance’ between any scale points and ii) limited redundancy and overlap between scale points.

The discussion of mechanisms in section 5 is in definitely in need of some updating with respect to the state-of-the-art. For instance, Brown et al. is a classic but may not cover some more recent instruction aspects that precisely play a role in answering incorrect Prompts; and Bender et al. is probably more critical than explanatory.
The section should better distinguish what depends on system Prompting and post-training from specific abilities (for instance the detection of logical inconsistencies may be predicated on ‘reasoning’ ability as well). There also needs to be a discussion, especially with the choice of completion parameters made (temperature) on the relationship between ‘creativity’ and tolerance to ill-formed Prompts, as some example from the paper would suggest. Finally, the last sentence of 5.2 which provides a tentative interpretation would need to be supported by stronger evidence and analysis.

Despite mentioning sensitive applications such as law and medicine in early sections of the paper, the actual examples appear to lack originality and contain standard disclaimers that are part of straightforward alignment, which is not informative neither supports the experimental plan.

The choice made of using limited-length Prompts may be more restrictive than initially intended as contradiction for instance is going to be almost formulaic rather than resulting from incompatibility between meaning of various sections of the Prompt. How this impacts on alignment especially with instructional tuning in mind would require better investigation.

Limitations are properly identified by the authors who offer a transparent discussion of their potential impact and how to mitigate them.
One limitation that might be considered is the paradox in testing short Prompts meant for a conversational use under an API LLM calling paradigm; the reason being that while it explores individual LLM differences in training and post-training as well as the steering ability of the system Prompt, it falls short of exploring the additional, albeit not always documented, steps that are involved in the consumer ‘chat’ form of many LLM implementations. There should be a discussion on whether additional ‘layers’ including user modeling or some form of dialog persistence or memory might impact clarification requests on incomplete statements.

---

> ### Author Response · Authors · 2026-07-01
> **Response to Reviewer gLgV**
>
> Thank you for these constructive comments. Each requested change is addressed below, with the corresponding revisions in the manuscript.
>
> **Taxonomy validation.** A new empirical validation subsection (Section 5.4) was added. Using classifier–human substitutions on a 315-item sample as a separability measure, the confusions are shown to concentrate on two boundaries (Pure Clarification vs. Clarification Plus; Terminal vs. Clarifying Refusal), which turn on graded quantities rather than sharp distinctions. The paper now states plainly that the taxonomy is not uniformly valid: it has a well-separated core and finer distinctions that lie on continua, so the even-distance property does not hold across all ten points. Consequently all quantitative claims are computed at the validated three-bucket level (over-compliant, clean, terminal; κ = 0.58 vs. 0.36 for the fine scheme), with the fine taxonomy retained only as descriptive.
>
> **Mechanisms (Section 5.2).** The discussion was revised to separate three contributions: preference-based post-training (linked to the sycophancy and reward-structure literature rather than to a generic next-token argument), the safety–helpfulness tension and its mirror image over-refusal, and, per your point, an explicit separation of response policy from the capability to detect ill-formedness—recognizing a contradiction plausibly draws on reasoning ability, and the near-ceiling contradiction/nonsense rates are consistent both with a policy favouring answers and with limits in detection, which the data cannot fully separate. A note on temperature and creative reinterpretation of nonsense was added and marked as suggestive, since temperature was not varied systematically; the previously tentative interpretation was softened accordingly.
>
> **Law/medicine examples.** These are now treated as one of the clean response categories rather than as a novel finding; the contribution rests on the cross-model comparison and the taxonomy, not on the disclaimers.
>
> **Limited-length prompts.** The limitations now acknowledge that short prompts make contradiction relatively formulaic, and identify longer, semantically embedded contradictions as future work.
>
> **Deployment layers (memory, user modelling).** A new limitation paragraph states that the API-level design characterises the model's response policy, not the product's, and that consumer chat layers—scaffolding, retrieval, user modelling, persistent memory—may request clarification or supply context before the model's disposition is expressed; replaying the benchmark through a deployed interface with and without memory is identified as a concrete extension.

---

> ### Comment · Reviewer_gLgV · 2026-07-21
> **Most concerns addressed**
>
> Thanks for this articulate and compact response.
> I have one remaining, albeit minor, concern in 5.2. I am moderately comfortable with a criticism of next-token prediction with references dating back 5+ years (p. 10).
> I would strongly suggest to address this in the paper's interest, although not making it a condition for positive recommendation.
> Also, I'm not sure this should be part of the paper "Reviewer feedback asked for validation of the taxonomy’s internal distinctness and overlap."